# Water and Nutrient Recovery for Cucumber Hydroponic Cultivation in Simultaneous Biological Treatment of Urine and Grey Water

**DOI:** 10.3390/plants12061286

**Published:** 2023-03-12

**Authors:** Anna Wdowikowska, Małgorzata Reda, Katarzyna Kabała, Piotr Chohura, Anna Jurga, Kamil Janiak, Małgorzata Janicka

**Affiliations:** 1Department of Plant Molecular Physiology, Faculty of Biological Sciences, University of Wrocław, Kanonia 6/8, 50-328 Wrocław, Poland; 2Faculty of Life Science and Technology, Wroclaw University of Environmental and Life Sciences, St. C. K. Norwida 27, 50-375 Wroclaw, Poland; 3Faculty of Environmental Engineering, Wroclaw University of Science and Technology, Wybrzeże Wyspiańskiego 27, 50-370 Wroclaw, Poland; 4Wroclaw Municipal Water and Sewage Company, Na Grobli 19, 50-421 Wroclaw, Poland

**Keywords:** cucumber, hydroponics, urine, grey water, salt stress, Na^+^/K^+^ balance, mineral deficiency

## Abstract

Water and nutrient deficiencies in soil are becoming a serious threat to crop production. Therefore, usable water and nutrient recovery from wastewater, such as urine and grey water, should be considered. In this work, we showed the possibility of using grey water and urine after processing in an aerobic reactor with activated sludge in which the nitrification process takes place. The resulting liquid (nitrified urine and grey water, NUG) contains three potential factors that can adversely affect plant growth in a hydroponic system: anionic surfactants, nutrient deficits, and salinity. After dilution and supplementation with small amounts of macro- and micro-elements, NUG was suitable for cucumber cultivation. Plant growth on this modified medium (enriched nitrified urine and grey water, NUGE) was similar to that of plants cultivated on Hoagland solution (HS) and reference commercial fertilizer (RCF). The modified medium (NUGE) contained a significant amount of sodium (Na) ions. Therefore, typical effects of salt stress were observed in cucumber plants, including reduced chlorophyll levels, slightly weaker photosynthesis parameters, increased H_2_O_2_ levels, lipid peroxidation, ascorbate peroxidase (APX) activity, and proline content in the leaves. In addition, reduced protein levels were observed in plants treated with recycled medium. At the same time, lower nitrate content in tissues was found, which may have resulted from their intensive use by nitrate reductase (NR), the activity of which significantly increased. Although cucumber is a glycophyte, it grew very well in this recycled medium. Interestingly, salt stress and possibly anionic surfactants promoted flower formation, which in turn could positively affect plant yield.

## 1. Introduction

The growing human population, global warming, increasing water crisis, and depletion of natural resources make nutrient solutions based on waste streams an alternative to conventional farming, which requires significant amounts of water, expensive mineral fertilizers, and detrimental pesticides. In recent years, the concept of wastewater reuse in agriculture has increased worldwide. The use of wastewater can be achieved through fertigation practices or through soilless cultivation systems [1,2,3]. A detailed description, as well as advantages, disadvantages, and risks, of fertigation can be found in [1,2]. Current studies have shown that human-derived urine and grey water, as essential components of wastewater, can be excellent sources of plant nutrient solutions [4]. Urine contains essential macro-elements for plant life, such as nitrogen (N), phosphorus (P), and potassium (K), which contribute 79%, 47%, and 71%, respectively [5]. It also comprises necessary microelements, such as B, Cu, Zn, Mo, Fe, Co, and Mn. A certain problem may be the high sodium (Na) content, which plants do not need. Additionally, urine contains a low concentration of heavy metal and small amounts of various intestinal microorganisms that can be easily disinfected [4]. Macro- and micronutrients are necessary for proper plant growth, creating a strong synergy between the developed recycling processes and plant cultivation. Unprocessed urine left in the environment contributes to the eutrophication of water; however, if used properly, it will become important for crop production [3]. Therefore, it is necessary to explore the possibility of using urine-based fertilizers for plant cultivation. However, the high ammonium concentration, high pH, and salinity of fresh urine can be toxic to plants. In general, the concentration of NH_4_^+^ above 0.1 to 0.5 mmol/L induced ammonium toxicity symptoms in plants; however, it can differ widely among plant species [6]. In addition, the availability of most nutrients is reduced in plants grown under alkaline conditions. High levels of salts, such as sodium, in raw urine (about 2.6 g/L) can have a negative effect on plant growth and thus yield [5]. Moreover, raw human urine has a high sodium adsorption ratio (SAR) of approximately 8.0. It is approximately 10 times more than that for tap water or soil [6]. For these reasons, raw urine should be properly treated before being used for plant cultivation. Urine nitrification allows urine stabilization, and ammonium changes to nitrate, which is a form of N that is safer for plants. Biological conversion of urine into a nitrate-rich solution is widely used in terrestrial wastewater treatment plants and is usually carried out in an activated sludge reactor that contains a substantial amount of nitrifying bacteria [3]. Nitrifying bacteria convert ammonia into nitrate, which is the primary source of N for plants. Simultaneously, during this process, precipitated P can be re-dissolved as a result of the pH decrease during nitrification [5].

Raw urine contains high concentrations of N and Na that are toxic to plants; therefore, its dilution is required. Grey water, as a larger source of recycled water, is one of the candidates used for the dilution of nitrified urine streams. Grey water is domestic wastewater that comes from sources such as dish washing, laundry, or bathing streams [7]. The main contamination of grey water is the accumulation of surfactants originating from detergents [8,9]. The impact of surfactants on plant growth has already been studied in soil and soilless types of cultivation [9,10]. In general, the effect of exposure to surfactants depends on their type and concentration. High concentrations cause significant limitations in plant growth, including plant death. In Janiak et al. [8], lettuce (*Lactuca sativa* var. *capitata* L.) cultivated in aeroponics and exposed to high concentrations (~2 g/L) of Sodium Laureth Sulfate (SLES), Sodium Methyl Cocoyl Taurate (SMCT), and Sodium Dodecylbenzenesulfonate (SDBS) resulted in a drastic decrease in yield and plant death. Similar results were obtained by Sawadogo et al. [9]. Growth of both okra (*Abelmoschus esculentus*) and lettuce was limited when exposed to commercial laundry detergent at a concentration of >1 g/L, and lethal in the case of higher concentrations (>5 g/L). However, low concentrations of surfactants had a moderate or little limited impact on the plants. In *Hydrocharis dubia* hydroponic cultivation, Linear Alkylbenzene Sulfonate (LAS) had a slight toxic effect on plant growth in the range of 0.5–20 mg/L [11]. Nevertheless, some surfactants can be highly biodegradable and delicate to plants, and they do not limit the use of grey water for cultivation [7]. 

Among the different types of plant cultivation, a hydroponic system seems to be an applicable choice for plant growth in nutrient solutions based on grey water and nitrified urine. Soilless culture systems are alternatives to conventional soil-based farming practices. The negative effects of soil type cultivation have been strongly discussed in recent years [12]. Among the main objections to conventional agriculture, the high and ineffective use of water, large land requirements, fertilizer use, soil degradation, and a decrease in biodiversity should be mentioned. Hydroponics are mainly intended for use in greenhouses. This can be important when space for growing crops is limited, such as in urban areas or environments with the risk of little fertile or contaminated soil [7,12]. However, the most important arguments in favor of hydroponics are the high yield, reduction in water consumption, high control of nutrient solution composition, and maintenance of stable plant growth conditions, as well as round-the-year production or minimal disease and pest incidence [7,13]. These all give us the opportunity to reach the healthiest plants with higher yields and better quality compared to traditional cultivation.

In hydroponic systems, plants such as cereals, grass, vegetables, fruit plants, fodder crops, flower plants, spices, and medicinal plants can be grown with better quality [12]. To date, various vegetables have been successfully grown under hydroponic conditions. Plants such as lettuce, spinach, or parsley—the leafy greens—can be cultivated in hydroponics easily with faster growth as a result of a shortened life cycle. Hydroponics can be used in the case of fruit crops, such as strawberries and blueberries, or plants that are sensitive to stress factors, such as drought (chives) or salinity (cucumber). Many hydroponic platforms are suitable and stable for larger plants, such as tomato and pepper [13]. Additionally, hydroponics have been used to investigate soil and water contamination caused by rare earth elements (REEs) in short-rotation crops (such as willow) [14], or in the case of grass (common reed) and ornamental flowers (iris) to examine the effect of Cd on plant growth [15]. The high potential of hydroponic cultivation is unquestionable, although it has certain limitations. One of them is technical knowledge and a repeatedly high investment. To optimize the cost of hydroponic production, especially water, its efficiency can be revised by recycling waste streams. Recycled urine has been directly applied to hydroponic systems in the case of lettuce [3,16], tomato [17] and dwarf tomato [18], as well as vegetables, such as amaranth and jute mallow [19]. The aim of this study was to investigate the possibility of using nutrient solutions based on recycled urine–gray water for hydroponic cucumber cultivation. To date, human urine has only been tested as a fertilizer for soil-type cultivation of outdoor cucumbers grown in a Nordic climate [20]. To the best of our knowledge, no experimental studies have been conducted on the effects of urine–gray water on cucumber soilless cultivation. 

In this study, we selected cucumber (*Cucumis sativus* L.) as a vegetable because of a few arguments. Cucumber is an economically important crop [21]. It is commonly cultivated throughout the world as the third-most produced vegetable. In addition, as a crop plant, cucumber is exposed to environmental stressors throughout its entire developmental life cycle, leading to limits in its yield and quality [22]. Cucumber has been successfully grown in hydroponic systems [13], so the conditions of its cultivation in hydroponics are well known and optimized. Finally, cucumber is a model plant used in our earlier studies concerning many aspects of the molecular mechanisms of cucumber adaptation to abiotic stress conditions, including salt stress, nutrient deficiency, extreme temperature, and heavy metals [23,24,25,26,27,28,29,30,31,32,33]. It is worth emphasizing that our previous experiments on cucumber were conducted in hydroponic cultivations [23,24,25,26,27,28,29,30,31,32,33]. These detailed characteristics of the cucumber provide an excellent biological reference point for studying the possibility of its growth under specific new conditions.

Therefore, we analyzed the use of a purified stream of nitrified urine and grey water, as well as a mixture based on nitrified urine and grey water with enrichment of missing macro- and microelements, in the hydroponic cultivation of cucumber (*Cucumis sativus* L.). We also used Hoagland’s solution as a standard hydroponic nutrient solution, and fertilizer for commercial vegetable cultivation as a reference. We verified and compared the effects of these solutions on cucumber growth parameters, mineral composition, protein content, photosynthetic pigment level, and chlorophyll fluorescence parameters. We determined the levels of nitrate and nitrate reductase (NR) activity in the cucumber tissues. Finally, we examined whether tested nutrient solutions based on nitrified urine and grey water cause stress in cucumber plants through determination of several oxidative stress parameters.

## 2. Results

### 2.1. Cucumber Response to NUG Solution

Cucumber seedlings displayed visible changes in the growth rate after the first days of growth in the NUG medium. NUG-cultivated plants showed strongly reduced growth when compared to NUGE-, HS-, and RCF-cultivated plants. After ten days of following NUG cultivation, the growth of the plants was completely inhibited, and the plants were characterized by strong chlorosis on the leaves (Figure 1), whereas after the next few days, leaf withering was observed. Consequently, the cultivation of the NUG plants ended around the 20th day of growth during the experiment. At the same time, no similar symptoms were observed in other plants nourished with NUGE, HS, and RCF media.

The results suggest that cucumber is unable to grow on NUG solution because of the deficiency of important macro- and micro-elements, an excessive amount of Na, and the high pH of NUG medium (Table 1), which can be reflected by the mineral composition of the NUG-cultivated cucumbers (Appendix A). NUG-nourished cucumbers had a decreased content of almost all analyzed nutrients (N, P, K, S, Ca, Mg, Fe, Mn, Cu, Zn, and B) and a high Na level measured in whole NUG-cultivated plants. Simultaneously, a slight correction of the composition and pH value of the NUG nutrient solution, which occurred in the case of the NUGE medium, resulted in improved growth, as well as elemental composition of cucumbers (Table 2). For these reasons, the remaining analyses presented in this paper were carried out only on cucumber plants capable of growing on the NUGE, HS, and RCF media.

### 2.2. Yield, Water Status, and Protein Content of Cucumber Plants

The yield of NUGE-, HS-, and RCF-cultivated cucumbers was determined after 30 days of growth and expressed as fresh and dry weights (DW) of the roots and shoots. The fresh mass of shoots of cucumber plants nourished with NUGE was similar to that of HS-cultivated plants and slightly lower than that of RCF-cultivated plants. The fresh mass of roots of NUGE-cultivated cucumbers was reduced by approximately 28% when compared to the RCF-growing plants (Figure 2A). The dry mass of the roots and shoots of plants nourished with NUGE did not change when compared to the RCF-cultivated plants (Figure 2B). Flower organ development was observed in 80% of cucumbers grown in NUGE solution, 83% of plants cultivated in HS, and only in 40% of RCF-cultivated plants. 

Among the common physiological responses of plants under environmental stress, water relations should be considered [34]. For this reason, the relative water content (RWC) was examined in the leaf tissue of NUGE-, HS-, and RFC-nourished cucumbers. The RWC of NUGE-cultivated plants decreased slightly by approximately 6% compared to the RCF-cultivated plants (Figure 2C). However, in all analyzed plants, the RWC level was high (above 90%), indicating that differences in the composition of the nutrient solutions did not significantly affect leaf water status and, thus, the ability of cucumber to absorb water.

Cultivation of the cucumber on the NUGE medium resulted in a decrease in soluble protein content. The concentration of protein in the leaves of NUGE-grown plants was approximately 15% lower than that in the leaves of plants grown in HS and RCF medium, where the soluble protein was at a similar level (Figure 2D).

### 2.3. Mineral Composition of Cucumber

The composition of the main macro- and microelements of leaves, stems, and roots of NUGE-, HS-, and RCF-cultivated cucumber plants is summarized in Table 2. The NUGE medium decreased the mineral content in leaves, stems, and roots of cucumber plants, except the Na content. The biggest differences (over 20%) were identified in leaves with respect to N (29%), K (over 36%), Ca (26%), Mg (27%), Mn, (22%), Cu (23%), Zn (24%), and B (26%), in roots in the content of P (35%), S (29%), Mn (35%), and Cu (35%), and in stems in the case of P (27%), Mg (29%), Mn (35%), Cu (31%), and Zn (21%) with regard to RCF-grown plants (Table 2). An excessive amount of sodium ions was observed in the leaves, roots, and stems of NUGE-cultivated plants compared to RCF-grown plants, at approximately 40%, 250%, and 190%, respectively (Table 2).

### 2.4. NR Activity and NO_3_^−^ Content

Tissue levels of nitrate ions and NR activity were determined in the cucumber leaves. Nitrate ions are the most important form of inorganic N that is taken up and assimilated by plants. They must be reduced through a two-step reduction to ammonium before incorporation into organic carbon skeletons to produce amino acids. NR located in the cytoplasm catalyzes the first step of nitrate reduction [35]. We found that the amount of tissue nitrate was significantly reduced in the leaves of NUGE-grown cucumbers and reached only 70% compared to the NO_3_^−^ level in reference to RCF-grown plants, where the highest level of NO_3_^−^ was observed. In plants growing on HS medium, the nitrate concentration in leaf tissue was also lower, approximately 10% compared to RCF plants (Figure 3A). At the same time, it was shown that NR activity reached the highest value in the leaves of plants growing on the NUGE medium. This accounted for approximately 172% of the NR activity detected in the leaves of the RFC-grown plants. NR activity measured in the leaves of HS-cultivated plants was also higher and reached 147%, compared to the enzyme activity in RCF-cultivated plants (Figure 3B).

### 2.5. Photosynthetic Pigments and Photosynthetic Activity 

Photosynthetic apparatuses are highly sensitive to changes in the environmental conditions to which plants are subjected [36]. The composition of the growing medium based on urine and grey water, as well as other factors, including nutrient element deficiency and/or excess sodium ions, may affect photosynthetic activity in plants [7]. Therefore, the concentrations of photosynthetic pigments and chlorophyll fluorescence parameters were determined in the leaves of cucumber plants grown on NUGE, HS, and RCF media. It was found that the chlorophyll *a* content was slightly reduced in the leaves of NUGE- and HS-cultivated plants compared to Chl a in RCF leaves (Figure 4). The amounts of chlorophyll b and carotenoids in NUGE-grown plants were also slightly reduced compared to those in RCF plants. 

Potential photosynthetic activity was evaluated using chlorophyll fluorescence transient techniques with JIP test analysis of parameters, which is a fast, non-invasive, and sensitive method [37,38]. Among the measured chlorophyll fluorescence parameters, the minimal fluorescence (Fo), maximal fluorescence of Chl a (Fm), and variable fluorescence related to the maximum capacity for photochemical quenching (Fv) were slightly lower in NUGE-grown plants than in RCF-grown plants (Table 3). Additionally, these parameters were similar in HS- and RCF-cultivated plants. Moreover, the time at which the maximum fluorescence value was reached (Tm) was longer in plants grown in the NUGE medium than in the HS- and RCF-grown plants. The area parameter indicated that the pool size of the electron acceptor Qa on the reducing side of PSII was reduced in leaves of NUGE-grown plants, and slightly increased in leaves of HS-grown plants compared to the value of this parameter measured in plants cultivated on RCF medium (Table 3). However, this did not significantly affect the efficiency of the water-splitting complex (Fo/Fm) and the maximum quantum efficiency of PSII (Fv/FM), which were unchanged in NUGE-, HS-, and RCF-cultivated cucumbers. This indicates that the capacity of PSII for photochemical quenching of energy within PSII and the physiological state of the photosynthetic apparatus were similar in the studied plants. The NUGE medium did not significantly affect the photosynthetic behavior of PSII compared to the HS and RCF media. The parameters describing the energy fluxes per active reaction center (RC), such as absorption of photon fluxes (ABS/RC), electron trapping (Tro/RC), and electron transport fluxes (Eto/RC), remained unchanged in plants cultivated on NUGE, HS, and RCF nutrient solutions. The dissipation of energy flux per cross section (Dio/CSo) was slightly reduced in NUGE-grown plants compared with RCF-grown plants. Although some of the measured parameters showed no differences in plants grown on the tested media, the performance index for energy conservation from photons absorbed by PSII and excitation on intersystem electron acceptors (PIABS), which can be used as an indicator of sample vitality, was slightly reduced in NUGE plants and slightly increased in HS plants compared to the reference RCF-grown plants (Table 3).

### 2.6. Analysis of Oxidative Stress Parameters

The excess of unfavorable sodium ions can induce oxidative stress in plant cells. For this reason, the H_2_O_2_ content was measured in the leaves of cucumber plants grown on all tested media. It was observed that both HS- and NUGE-cultivated plants accumulated more H_2_O_2_ than plants nourished with RCF. However, the highest H_2_O_2_ levels were determined in the leaves of NUGE-nourished cucumbers, reaching about 145% and 122% in comparison to the RCF- and HS-cultivated plants, respectively (Figure 5A).

One of the indicators of oxidative stress in plant tissues is enhanced lipid peroxidation. It was found that the level of lipid peroxidation increased in the tested plants in parallel with the changes in the H_2_O_2_ content. The highest amounts of TBARS were shown in the leaves of the NUGE-cultivated plants. They achieved about 161% and 126% comparing to the leaves of RCF- and HS-nourished plants, respectively (Figure 5B).

To minimize hydrogen peroxide accumulation, plants activate the H_2_O_2_-decomposing enzymes, primarily catalase (CAT) and ascorbate peroxidase (APX). In cucumber leaf tissues, CAT activity was maintained at a constant level in all plants, regardless of the composition of the nutrient solution (Figure 5C). However, APX activity was slightly stimulated in the leaves of cucumbers grown on the NUGE medium compared to those grown on RCF and HS (Figure 5D).

Under various stress conditions, proline acts not only as an osmolyte but also as an antioxidant defense molecule. The level of proline changed significantly in the cucumber leaves depending on the medium used. It reached the highest value in plants grown on NUGE and was the lowest in plants grown on HS. The proline content of NUGE-nourished plants was about 45% and 16% higher than in HS- and RCF-cultivated plants, respectively (Figure 5E).

## 3. Discussion

For several years, there has been an increasing amount of discussion regarding the deepening water deficit crisis [7]. With growing demands from human activities on the one hand, and climate change problems on the other, many regions around the world are struggling to provide enough freshwater. It is important to take measures to ensure the continuity of food production without exposing the environment to a shortage of fresh water. One of the actions in this area is to improve water use efficiency. This can be obtained by recycling water and elements from urine and grey water. Grey water is an abundant source of water, while urine can be a major source of some macro- and microelements that are necessary for plant growth [4,5]. The biological conversion of urine into a nitrate-rich medium diluted with grey water is a candidate for plant fertilizers [3].

Our previous studies have shown that the appropriate modification of grey water and urine allows the cultivation of lettuce [7]. In the present study, we demonstrated the use of grey water and urine in the hydroponic cultivation of cucumbers. According to Statista, in 2021, the global production of cucumber increased to over 93 million metric tons, making cucumber the third most-produced vegetable. The production of NUG in the nitrification reactor was conducted as described in [39]. NUG is not an ideal medium for plant growth, although it allows for the growth of lettuce [7], whereas cucumber is unable to grow properly on this medium. 

The product obtained from grey water and urine was characterized by a very high sodium ion content and low content of important macro-elements, such as K, P, Ca, and Mg, as well as microelements (Table 1). In addition, the pH of the reactor effluent was slightly higher than the optimal pH (7.2). This makes it more difficult for plants to absorb nutrients. For example, plants can consume calcium at a relatively low pH. The alkaline pH of the medium reduces the availability of most micronutrients and macronutrients, such as P. At a pH above 7.0, P reacts with calcium ions to form an insoluble calcium phosphate salt [40]. Lettuce coped with such unfavorable conditions [7], but cucumbers did not survive after a few days of NUG treatment. Cucumber seedlings suffered primarily from mineral deficiencies; however, as glycophytes, they found it difficult to grow in the presence of such large amounts of sodium ions and reduced potassium content. In the NUG medium, the ratio of potassium to sodium ions was extremely low (0.15). In the reference medium (RFC), this ratio reached 23.5. 

Media containing human urine are always high in sodium and chloride ions [18]. Moreover, during the nitrification process, Na ions were introduced as NaHCO_3_ to adjust the pH and serve as an alkalinity source for nitrifying bacteria [39]. Salt stress contributes to potassium–sodium imbalance. K is an important macronutrient. In NUG, its content was very low, and an excessive amount of sodium hindered K^+^ uptake by plants. Na^+^ uptake is largely mediated through channels and transporters, such as the high-affinity K^+^ transporter (HKT) and non-selective cation channels (NSCC) at the plasma membrane [41,42]. Hydrated sodium and potassium ions are similar in size, indicating that they can be transported by the same transporters. Therefore, if there are more Na^+^ ions in the environment, they compete with K^+^ ions for transporters. Such difficult conditions related to the composition of the NUG medium prevent cucumber seedlings from growing, causing them to wither and die. In addition, high concentrations of sodium and chloride cause salt stress, and cucumber is sensitive to salinity.

Supplementing the NUG medium with deficient nutrients (K, Mg, P, Ca, and microelements) and lowering the pH of the nutrient solution to 6.5 did not eliminate salt stress, but significantly minimized its effects. We found that the cucumber seedlings grew better on the enriched NUGE medium. The growth of these seedlings was comparable to that of plants cultivated on HS and RCF. The introduction of potassium (KH_2_PO_4_ and K_2_SO_4_) increased the K^+^/Na^+^ ratio from 0.15 in NUG to 0.55 in NUGE. Such a small change was beneficial for the potassium–sodium balance and, thus, the growth of plants (Table 2). The effects of salt stress can still be observed in plants but are not lethal to cucumbers. In addition, the pH of the NUGE medium was more favorable (lowered from 7.2 to 6.5), which facilitated the uptake of essential nutrients by the plants (Table 1).

Symptoms of salt stress were visible in plants grown in the NUGE solution. Plants grown on NUGE medium accumulated three times more sodium ions in the roots and stems than plants grown on a reference medium (Table 2). However, less sodium accumulated in the leaves than in the roots. There were only 1.4 times more sodium ions in the leaves of the NUGE plants than in the leaf tissues of plants grown on the reference medium. Sodium transport plays a major role in plant salinity tolerance. Plants must minimize the presence of toxic sodium ions in photosynthetic tissues (in the leaves) and growth meristems by sequestering it in older parts or in the roots. Moreover, plants store Na^+^ in vacuoles and cell walls (apoplast) to alleviate Na^+^ toxicity in the cytosol. Vacuolar and apoplast Na^+^ sequestration is mediated by Na^+^/H^+^ antiporters, SOS1 in the plasma membrane, and NHX in the tonoplast [41,43]. In the present study, the accumulation of Na^+^ ions in the roots was also observed (Table 2). Based on the *sos1* mutants of *Arabidopsis*, it has been suggested that SOS1 controls both Na^+^ efflux in the roots and long-distance Na^+^ transport via the xylem [44].

When Na^+^ enters the root endodermis, it is translocated to the shoot via the xylem. In plants subjected to salinity stress, the Na^+^ concentration in the xylem sap is often high [44]. In the present study, a high level of Na^+^ was observed in the stems of NUGE plants (more than three times higher than in plants not subjected to salinity) (Table 2). In tomatoes with reduced *SOS1* expression, enhanced sensitivity to salinity was shown. Salt stress causes an increase in the accumulation of Na^+^ in the leaves and roots but a lower content in the stems of plants with reduced *SOS1* gene expression. The ability of tomato stems to retain Na^+^ to prevent Na^+^ from entering photosynthetic tissues is significantly dependent on the functioning of SOS1 [45]. In the cucumber plants, the transport of Na^+^ to the leaves was significantly reduced (Table 2). This allowed the plants, despite being exposed to salt stress, to carry out the photosynthesis process, which was only slightly impaired when compared to the control plants.

Photosynthesis is one of the principal processes occurring in plant chloroplasts. Chlorophyll (Chl) is a significant component of the thylakoid membrane and reflects the photosynthetic capacity of plants. This component plays a major role in the capture and transfer of light energy during photosynthesis and is one of the most important indicators of salt tolerance. The chlorophyll a, b, and carotenoid contents were slightly reduced in the leaves of plants grown on NUGE compared to those of plants grown on RCF medium (Figure 4). These data are consistent with many previous reports indicating that salt stress reduces the levels of photosynthetically active pigments in plants that are sensitive to salinity [46,47,48]. The decrease in Chl content may be attributed to increased degradation and/or inhibition of pigment synthesis [49]. Salt stress can affect chloroplast structure and decrease chlorophyll content, resulting in a reduced photosynthetic rate [47]. In the present study, the Fo, Fm, and Fv were slightly lower in NUGE-grown plants than in RCF-grown plants. It has been reported that the maximum quantum yield of photosystem II (PSII; Fv/Fm) decreased in sorghum under saline conditions [50]. In poplar (*Populus* spp.), salt stress can inhibit Fv/Fm owing to the salt-induced increase in (Fo and decline in Fm [51]. Although the photosynthetic parameters were not significantly reduced, their disturbance could have affected the production of the carbon skeletons required for protein production.

Cultivation of cucumber on the NUGE medium resulted in a decrease in the soluble protein content (Figure 2D). Plant tissues respond to salt stress by degrading proteins or producing abundant salt stress-related proteins [52]. Soluble proteins accumulated in plants under salt stress play a key role in osmoregulation but also protect cell membranes, and enzymes from Na^+^ toxicity. More proteins have been observed in salt-tolerant cultivars than in the salt-sensitive cultivars of many crops. In *Paulownia imperialis* (a salt-sensitive plant), the soluble protein content decreases under salt stress conditions [53]. Cucumber is a salt-sensitive plant, which may explain why we observed reduced levels of soluble proteins. Potassium ions can also significantly affect the soluble protein content in plants. K^+^ regulates enzymes both transcriptionally and post-transcriptionally [54]. In this study, in the NUGE medium, despite the similar level of K^+^ to that in the RCF, sodium ions and their competition for carriers hindered the uptake of potassium ions by plants. The accumulation of K^+^ in the leaves of NUGE plants was almost half that of the leaves of plants grown on the reference medium. The K status of plants has an important effect on the transport and distribution of photosynthetic products [55]. The correct level of K^+^ ions is necessary to create osmotic potential in the phloem and to help transfer photosynthetic products from the leaves to other organs. Photosynthate loading in the phloem is inhibited in plants deficient in K^+^ [56]. This phenomenon may also have contributed to the reduction in the production of amino acids and proteins in plants grown on NUGE (Figure 2D). 

K^+^ deficiency can also affect plant water content. K^+^ plays a crucial role in turgor regulation in guard cells during stomatal movement. The rapid release of K^+^ from guard cells contributes to stomatal closure. Therefore, it seems that stomata find it difficult to remain open under K-deficient conditions [57]. However, many studies have shown that K starvation can favor stomatal opening under drought stress [58,59]. In plants with K^+^ deficiency, water stress-induced stomatal closure can be inhibited by ethylene synthesis [59]. K^+^ starvation stimulates ethylene production [59]. Increased ethylene accumulation can reduce the effect of abscisic acid (ABA) on stomata and delay stomatal closure [60]. Therefore, stomata cannot function properly in K^+^-deficient plants, which results in greater water loss. In the present study, a slight decrease in the relative water content of NUGE-cultivated plants (with a decreased amount of K^+^ in the leaves) was observed (Figure 2C).

Several studies have confirmed a relationship between K and N metabolism. K^+^ activates enzymes involved in plant growth, protein synthesis, and photosynthesis [61]. The acquisition rates of K^+^ and NO_3_^–^ are often positively correlated, and a sufficient K supply can promote N metabolism and enhance the synthesis of amino acids and proteins [61,62]. Moreover, K^+^ deficiency in plants triggers reductions of NO_3_ uptake [63]. In cotton, K^+^ deficiency can inhibit NO_3_^−^ absorption and significantly reduce NO_3_^−^ content in the leaves [64]. Similarly, in this study, a deficiency of K^+^ could be one of the reasons for the reduced level of nitrates in the tissues of cucumbers growing on NUGE (Figure 3A). However, the reduced nitrate content in tissues could also result from its intensive use by NR, the activity of which was significantly increased in cucumber plants grown on the NUGE medium (Figure 3B). Among the complex assimilation of NO_3_^−^, the rate-limiting step in this process is the reduction of nitrate to nitrite by NR. NR is an enzyme affected by salt stress [32,65,66,67]. We hypothesized that the increase in NR activity in plants grown on NUGE could be due to the salt stress to which the plants were exposed. In our previous study, higher NR activity was observed in NaCl-treated cucumber roots [32]. The increase in NR activity was due to the stimulation of *CsNR* gene expression and post-translational modifications, such as enzyme dephosphorylation [32]. The increase in NR activity under stress conditions (owing excess Na^+^ and K^+^ deficiency), during which plants have reduced protein levels and impaired photosynthesis, is important for mitigating the adverse effects of these conditions. The increased activity of NR contributes to the effective reduction of nitrates and, thus, to the production of ammonium ions, which can be used to produce amino acids and proteins. Therefore, this appears to be an essential element for cucumber adaptation to unfavorable conditions related to NUGE nutrition.

Salt stress and mineral deficiencies often lead to secondary oxidative stress. Plants adapt to salinity stress by activating enzymes that scavenge reactive oxygen species (ROS) and improve osmotic adjustment [52]. In cucumber plants grown on NUGE, we observed increased H_2_O_2_ levels, lipid peroxidation, APX activity, and proline content in the shoots (Figure 5). The reaction of plants to salinity is often accompanied by increased production of H_2_O_2_ [24,30,68,69]. NUGE-cultivated plants accumulated much more H_2_O_2_ than the plants nourished with RCF (Figure 5A). Lipid peroxidation is a biochemical marker of oxidative stress in plants [70]. The highest amount of TBARS was also observed in the leaves of the NUGE-cultivated plants (Figure 5B), which demonstrated increased peroxidation of lipids and, consequently, the induction of oxidative stress in these cucumber plants. On the one hand, hydrogen peroxide is a harmful substance that damages membranes and proteins, and thus disrupts many physiological processes. However, on the other hand, hydrogen peroxide is a signaling molecule necessary for the activation of plant defense mechanisms in response to environmental stress factors. It is important to balance ROS production with scavenging to protect plant cells against excessive ROS production [71]. The harmful effects of ROS in plants can be reduced by non-enzymatic and enzymatic defense systems. It has been reported that proline acts as a nonenzymatic antioxidant [72]. Many studies have shown that salt stress stimulates the expression of genes involved in proline biosynthesis, leading to proline accumulation [73,74]. In the present study, the level of proline changed significantly in cucumber leaves. It reached its highest value in plants grown on NUGE medium (high level of Na^+^). The enzymatic antioxidant system consists of CAT and APX, among others, which are activated in plants to break down H_2_O_2_ [75]. APX activity was slightly stimulated in cucumbers grown on NUGE (Figure 5D), whereas CAT activity did not change. Stimulation of both APX and CAT in response to NaCl has been reported in many plant species [30,76]. However, a greater increase in CAT activity was observed in salt-tolerant plants but not in salt-sensitive plants [77]. In some glycophytes, the activity of this enzyme decreases after NaCl treatment [78]. The cucumber we studied is a glycophyte sensitive to salinity. Therefore, it seems likely that no changes in CAT activity were observed in plants grown on NUGE, i.e., exposed to salt stress.

## 4. Materials and Methods

### 4.1. Cultivation of Plants

*Cucumis sativus* L. cv. Wisconsin seeds were germinated for 48 h in the dark and subsequently watered with 3× diluted Hoagland’s solution and transferred to a growing chamber for 5 days (temperature of 25 °C during the day, and 22 °C at night). After that, well-grown healthy seedlings were placed in 1.8 l containers (with 2 seedlings per container) with nutrient solutions continuously aerated and grown hydroponically for 30 days under a 16-h photoperiod, temperature of 25 °C during the day, and 22 °C at night. Four different variants of nutrient solutions were used for cucumber cultivation. NUG medium was based on diluted (3 times) Nitrified Urine and Grey water, pH 7.2. NUGE solution, pH 6.5, was also based on nitrified urine and grey water (diluted 3 times) and enriched with missing macro-elements (CaNO_3_, KH_2_PO_4_, K_2_SO_4_) and microelements (B, Cu, Fe, Mn, Mo, Zn). HS medium (Hoagland’s solution), pH 6.5, as a physiological medium, was the third option of the nutrient solution. The last one, RCF, pH 6.5, is Green Superba Yara, a fertilizer for commercial vegetable cultivation, used here as a control. The composition of all types of media used in the experiment is described in Table 1. The production process of urine and grey water was described in detail in [7,39]. A short description is also presented in the Appendix A. The experiment was repeated in two independent cultivations. After the cultivation period, the shoots and roots of each cucumber plant, at least six from each nutrient solution used, were separated and weighed. The plant material was dried at 80 °C for 48 h for DW determination.

### 4.2. Analytical Methods

The RWC was established according to the method of [79]. For this analysis, fully grown cucumber leaves were selected. To estimate the FW, six discs from each leaf were cut and weighed, placed into Petri plates with distilled water, and kept in the dark for 24 h. After that, the discs were reweighed to determine turgid weight (TW), and oven-dried at 80 °C for 48 h to designate DW. The RWC was estimated using the formula from González and González-Vilar [80]:RWC = ((FW − DW)/(TW − DW)) ∙ 100%

The nitrogen content in cucumber roots, leaves, and shoots was measured using the Kjeldahl method [81] with a Gerhard Vapo Dest 20 apparatus. The content of other macro- and microelements was determined after mineralization of the plant roots, leaves, and shoots in the Milestone Ethos UP mineralizer using the ICP technique. To determine the mineral composition of cucumbers from each variant (n = 5), leaf, root, or shoot materials were collected and mixed into an average sample. Each cucumber composition measurement was conducted four times.

The endogenous nitrate content was determined in aqueous extracts prepared from 100 mg of fresh leaf tissue powdered in liquid N, as described by [82]. The amount of NO_3_^−^ ions in the samples was measured using a HPLC system with a Sphere-Image 80-5 SAX ion exchange column (Knauer). The level of soluble protein was determined according to the standard protocol of Bradford [83], and bovine serum albumin (BSA) was used as a standard.

The photosynthetic pigment content was determined based on the method of Lichtenthaler and Wellburn [84]. The chlorophyll a and b and carotenoid concentrations were measured at a density of 80% (*v*/*v*) acetone extracts using a Beckman spectrophotometer at wavelengths of 646.8, 663.2, and 470 nm. To calculate the concentrations of individual levels of photosynthetic pigments, the following formulas based on [84] were used with some modifications:Chl a = 12.2 ∙ A_663.2_ − 2.79 ∙ A_646.8_(1)
Chl b = 21.50 ∙ A_646.8_ − 5.10 ∙ A_663.2_
(2)
Carot. = (1000 ∙ A_470_ − 1.82 ∙ Chl a − 85.02 ∙ Chl b)/198(3)

The chlorophyll *a* fluorescence parameters were determined using a Handy PEA + Chlorophyll Fluorimeter (Hansatech Instruments, Pentney, UK). The leaves were dark-adapted for 30 min using leaf clips to complete full re-oxidation of photosystem 2 (PS2), and the measurements were performed using a red light (650 nm) saturating pulse of 3500 µmol ∙ m^−2^ ∙ s^−1^, a pulse duration of 1s, and a fixed gain of 0.7. The fluorescence parameters were calculated using the OJIP test method [85]. The analyzed parameters are listed in Appendix A. PEA + software (Hansatech) was used as a comprehensive tool for in-depth analysis of the recorded data.

The lipid peroxidation level was assayed as described by [86] with some modifications. The method is based on the formation of thiobarbituric acid reactive substances (TBARS) in an acidic pH of 90–100 °C. Frozen leaf material was homogenized with 0.1% trichloroacetic acid (TCA) and centrifuged at 15,000× *g* for 5 min. The reaction mixture consisted of 0.3 mL of supernatant and 0.8 mL of 0.5% thiobarbituric acid (TBA) dissolved in 20% TCA. After 30 min of incubation at 95 °C, the samples were immediately cooled and centrifugated at 10,000× *g* for 10 min. The absorbance of the supernatant was measured at 532. The correction of nonspecific turbidity was made by subtracting the absorbance of the same at 600 nm. An extinction coefficient of ε = 155 mM^−1^ ∙ cm^−1^ was used. 

The total hydrogen peroxide content (H_2_O_2_) was quantified according to the method of [87] with modifications by [31]. The leaf tissue was ground into a fine powder in liquid nitrogen, homogenized with 0.1% TCA and centrifuged at 12,000× *g* for 20 min at 4 °C. The supernatant containing 10 mM K-phosphate buffer (pH 7.0) and 1 M KI was added to the reaction mixture. After incubation of the samples in the dark at room temperature for 60 min, the absorbance of the reaction product, triiodide (I_3_^−^), was recorded at 390 nm. 

CAT (EC 1.11.1.6) and APX (EC 1.11.1.11) activities were determined as previously described [30]. The leaf tissue was homogenized with 100 mM K-phosphate buffer (pH 7.5) containing 1 mM EDTA and 5 mM ascorbic acid. The homogenate was then filtered and centrifuged at 15,000× *g* for 10 min at 4 °C [88]. The supernatant obtained was used for CAT and APX determination. The CAT activity was examined according to the method described by Aebi [89], with modifications described by Janicka-Russak et al. [28]. The decomposition of H_2_O_2_ was followed by measuring the decrease in absorbance at 240 nm for 150 s at 30 s intervals and was calculated every 60 s. The reaction mixture consisted of 50 mM phosphate buffer (pH 7.0), plant extract, and 10 mM H_2_O_2_. One unit of CAT was defined as the amount of enzyme that breaks down 1 μmol of H_2_O_2_·min^−1^. The APX activity was determined in a mixture containing 100 mM potassium phosphate (pH 7.0), 0.5 mM ascorbate, 0.2 mM H_2_O_2_, and enzyme extract, according to the protocol of Chen and Asada [88]. H_2_O_2_-dependent oxidation of ascorbic acid was followed by monitoring the decrease in absorbance at 290 nm, assuming an absorption coefficient value of 2.8 mM^−1^ ∙ cm^−1^.

The NR (EC 1.6.6.1) activity was determined in the crude extract obtained from leaf tissue, as previously described by Reda [82]. The enzymatic reaction was conducted in 50 mM Hepes-KOH buffer (pH 7.5) in the presence of 10 mM KNO_3_ and 5 mM EDTA, according to Kaiser and Huber [90]. The amount of nitrite ions was measured colorimetrically using Griess reagent according to Reda et al. [32]. 

The proline content was measured using the acid-ninhydrin procedure described by Bates et al. [91] with some modifications. Leaf tissue (0.5 g) was homogenized in 5 mL of 3% sulfosalicylic acid and centrifuged at 10,000× *g* for 20 min at 4 °C. The clear supernatant (1 mL) was mixed with an equal volume of acid-ninhydrin and acetic acid reagent and incubated at 100 °C for 1 h. The reaction was stopped by cooling the sample in an ice bath. The proline content was measured colorimetrically after extraction of the reaction mixture with 2 mL of toluene. The absorbance was read at 506 nm with toluene as a blank. The amount of proline was calculated using a standard curve and is presented as the concentration of proline × g FW^−1^.

The physiological significance of some of the measured parameters is presented in the Appendix A.

### 4.3. Statistical Analysis

During the experiments, data from at least five plants were averaged, and the distribution of values in the sample was presented as the standard deviation. Each analysis was repeated at least five times. The results were analyzed using Student’s t-test (α = 0.05), and statistically significant differences between the test and control values are indicated with an asterisk in all figures and tables.

## 5. Conclusions

In summary, considering the presented results, we believe that appropriately modified grey water and urine can be successfully used for the hydroponic cultivation of cucumber plants. The tested NUGE nutrient solution contained significant amounts of sodium and chloride ions, which were related to the nitrified urine component of this medium. Excess toxic sodium ions contribute to salt stress symptoms in cucumbers. However, we noticed that the plants grew well and triggered defensive reactions, and their growth almost did not differ from that of cucumbers on reference nutrient solutions. The plants were properly nourished and contained a typical amount of minerals in their tissues. In addition, we observed that the plants grown in the NUGE medium had significantly more flowers than those grown in the optimal reference medium, which was probably due to salinity or the presence of surfactants. It seems interesting whether the observed increase in the number of flowers will have an impact on the higher yields and quality of cucumbers growing on NUGE. This aspect is important and requires further study. We intend to extend the time spent growing plants until the fruits are harvested. In addition, we would like to check the extent to which surfactants are an element affecting the growth of cucumber plants on NUGE. Do surfactants in the face of salinity matter for plant growth? With regard to the fertilizer production process by means of nitrification, the next steps will cover further optimization. 

## Figures and Tables

**Figure 1 plants-12-01286-f001:**
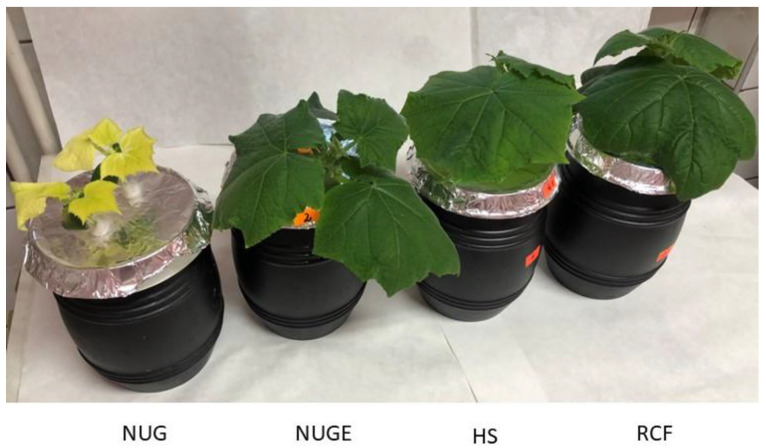
Cucumber plants cultivated on NUG, NUGE, HS, and RCF media.

**Figure 2 plants-12-01286-f002:**
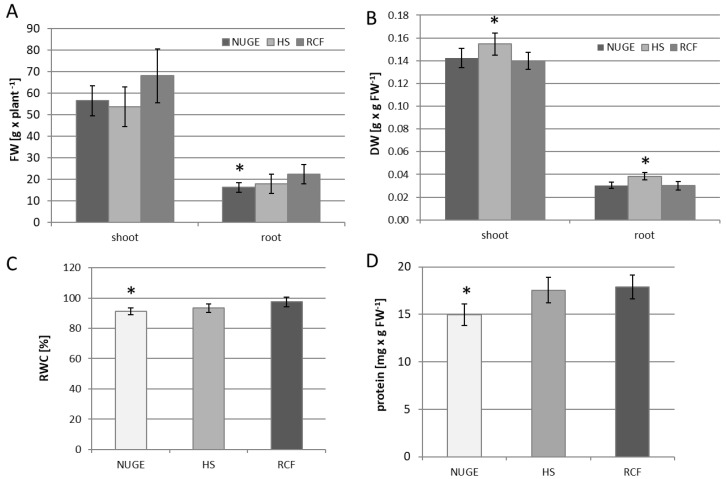
The different parameters of NUGE—, HS—, and RCF—nourished cucumbers obtained after 30 days of cultivation. (**A**)—Fresh weight (FW). (**B**)—Dry weight (DW) of shoots and roots. (**C**)—Relative water content (RWC). (**D**)—Protein level in cucumber leaves. The data ± SD are the means of at least n = 5 replicates. Asterisks indicate significant differences with *p* ≤ 0.05.

**Figure 3 plants-12-01286-f003:**
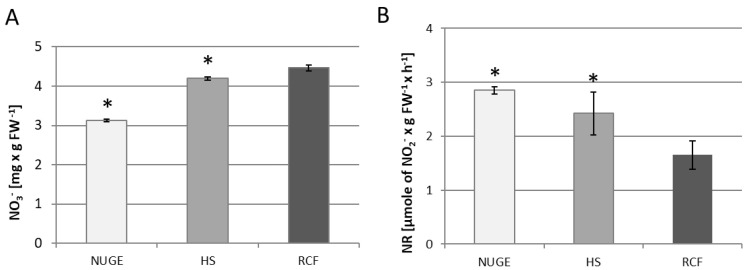
Nitrate ion content (**A**) and nitrate reductase (NR) activity (**B**) in leaves of cucumber plants grown on nitrified urine and grey water enriched with fertilizers, NUGE; Hoagland nutrient solution, HS, and Referenced Commercial Fertilizer, RCF media. Tissues were collected after 30 days of cultivation. Analyses were performed as described in the Material and Methods. The presented data ± SD are the means of at least n = 5 replications. Asterisks indicate significant differences (*p* ≤ 0.05).

**Figure 4 plants-12-01286-f004:**
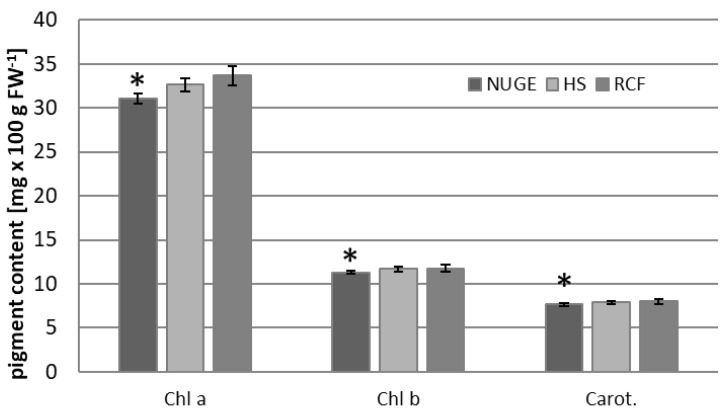
Photosynthetic pigment (Chlorophyll a, Chlorophyll b, and Carotenoids) contents in the leaves of cucumbers grown on nitrified urine and grey water enriched with fertilizers, NUGE; Hoagland nutrient solution, HS, and Referenced Commercial Fertilizer, RCF media. The presented data ± SD are the means of at least n = 5 replicates. Asterisks indicate significant differences at *p* ≤ 0.05.

**Figure 5 plants-12-01286-f005:**
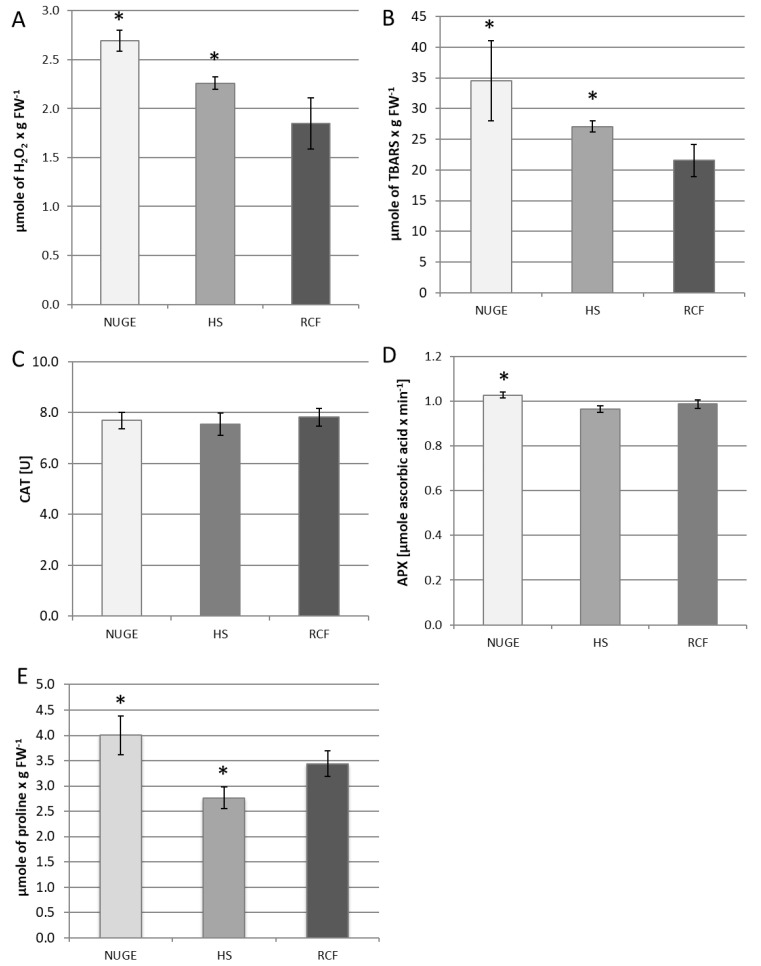
Effect of different plant nutrition on parameters related to oxidative stress. Leaves were collected from cucumber plants grown on NUGE, HS, and RCF media for 30 days. (**A**)—H_2_O_2_ content. (**B**)—Lipid peroxidation. (**C**)—Catalase (CAT) activity. (**D**)—Ascorbate peroxidase (APX) activity. (**E**)—Proline level. All parameters were determined as described in the Material and methods. The presented data ± SD are the means of at least n = 5 replicates. Asterisks indicate significant differences at *p* ≤ 0.05.

**Table 1 plants-12-01286-t001:** Compositions of NUG, NUGE, HS, and RCF used in the experiment. The pH of NUGE, HS, and RCF media was regulated up to 6.5-fold using 1M NaOH.

Element	Unit	NUG	NUGE	HS	RCF
pH	-	7.21	6.5	6.5	6.5
N-NH_4_^+^	mg·L^−1^	trace	trace	trace	trace
N-NO_3_^−^	mg·L^−1^	189	192	224	180
P-PO_4_	mg·L^−1^	20	41	62	60
K	mg·L^−1^	73	265	235	250
Ca	mg·L^−1^	31	150	160	150
Mg	mg·L^−1^	7	25	24	50
Na	mg·L^−1^	497	485	12.5	10.6
Cl	mg·L^−1^	205	198	1.77	3.7
S-SO_4_^2−^	mg·L^−1^	166	208	32	30
Fe	mg·L^−1^	trace	1.5	2.00	2.00
Mn	mg·L^−1^	trace	0.75	0.80	0.75
Cu	mg·L^−1^	trace	0.05	0.03	0.10
Zn	mg·L^−1^	trace	0.35	0.13	0.50
B	mg·L^−1^	0.07	0.35	0.27	0.40
Mo	mg·L^−1^	trace	0.05	0.05	0.05
Anionic surfactant SMCT	mg·L^−1^	31	26	0	0

**Table 2 plants-12-01286-t002:** Mineral composition of NUGE-, HS-, and RCF-cultivated cucumber organs. Number of cucumber plants of each medium variant was n = 5. All parameters in the analyzed variants (NUGE and HS) differed from those obtained in the reference (RCF) solution in a statistically significant way (*p* ≤ 0.05).

Nutrient solution	Cucumber organ	Macroelements in g·kg DM^−1^					
		N	P	K	Ca	Mg	S
NUGE	leaf	30.5 ± 0.24	5.17 ± 0.12	26.5 ± 0.3	19.6 ± 0.36	3.2 ± 0.08	4.47 ± 0.25
HS		47.2 ± 0.25	7.83 ± 0.12	44.5 ± 0.2	30.2 ± 0.2	4.8 ± 0.1	6.2 ± 0.1
RCF		43.1 ± 0.21	6.17 ± 0.26	41.2 ± 0.3	26.4 ± 0.2	4.4 ± 0.1	5.5 ± 0.08
NUGE	root	14.5 ± 0.12	2.47 ± 0.05	15.5 ± 0.02	3.2 ± 0.05	3.4 ± 0.05	1.2 ± 0.1
HS		16.5 ± 0.12	4.23 ± 0.05	15.2 ± 0.2	4.1 ± 0.1	3.8 ± 0.08	1.4 ± 0.05
RCF		15.2 ± 0.22	3.8 ± 0.08	16.5 ± 0.3	3.6 ± 0.1	4.1 ± 0.08	1.6 ± 0.09
NUGE	stem	18.9 ± 0.12	4.53 ± 0.12	10.5 ± 0.2	9.2 ± 0.1	2.8 ± 0.1	4.8 ± 0.1
HS		21.2 ± 0.24	5.6 ± 0.16	11.2 ± 0.2	12.1 ± 0.2	4.23 ± 0.1	5.2 ± 0.1
RCF		22.4 ± 0.17	6.17 ± 0.2	11.6 ± 0.12	11.2 ± 0.2	3.9 ± 0.1	4.3 ± 0.1
Nutrient solution	Cucumber organ	Microelements and sodium in mg·kg DM^−1^					
		Fe	Mn	Cu	Zn	B	Na
NUGE	leaf	74.47 ± 0.74	75.60 ± 0.41	6.25 ± 0.05	48.57 ± 0.39	46.23 ± 0.45	845.3 ± 9.3
HS		121.47 ± 0.4	102.27 ± 0.3	9.54 ± 0.04	56.53 ± 0.4	72.13 ± 0.31	631 ± 7.1
RCF		89.63 ± 0.34	96.5 ± 0.29	8.14 ± 0.01	64.23 ± 0.34	62.13 ± 0.25	598.3 ± 2.1
NUGE	root	86.47 ± 0.26	55.57 ± 0.12	32.13 ± 0.21	86.63 ± 0.26	82.33 ± 0.17	1123.3 ± 4.5
HS		106.3 ± 0.36	98.63 ± 0.21	45.63 ± 0.37	102.33 ± 0.4	72.43 ± 0.12	367.7 ± 1.2
RCF		98.63 ± 0.09	85.23 ± 0.12	49.80 ± 0.16	92.13 ± 0.12	80.23 ± 0.31	322 ± 2.4
NUGE	stem	86.23 ± 0.66	62.5 ± 0.22	17.54 ± 0.12	72.40 ± 0.29	56.77 ± 0.37	1322 ± 14.5
HS		98.63 ± 0.42	85.57 ± 0.26	28.6 ± 0.37	87.46 ± 0.73	76.33 ± 0.56	421.3 ± 6.9
RCF		92.3 ± 0.36	96.3 ± 0.41	25.57 ± 0.5	92.27 ± 0.48	71.17 ± 0.37	452.3 ± 3.3

**Table 3 plants-12-01286-t003:** Effect of NUGE–, HS–, and RCF–nutrient composition on cucumber’s photosynthetic activity analyzed by the chlorophyll *a* fluorescence method (OJIP test). The presented data ± SD are the means of at least n = 5. Asterisks indicate significant differences (*p* ≤ 0.05). NUGE, enriched nitrified urine and grey water, HS, Hoagland solution, RCF, reference commercial fertilizer.

	NUGE ± SD	HS ± SD	RCF ± SD
T for Fm	343 ± 89	312 ± 44	314 ± 48
Area	55,364 ± 5741	61,749 ± 4346	58,984 ± 4536
Fo	425 ± 12	448 ±20	457 ± 21
Fm	2595 ± 131 *	2679 ± 56	2676 ± 42
Fv	2169 ± 125	2238 ± 77	2229 ± 29
Fo/Fm	0.164 ± 0.007	0.165 ± 0.01	0.167 ± 0.009
Fv/Fm	0.836 ± 0.007	0.839 ± 0.009	0.833 ± 0.009
ABS/RC	2.064 ± 0.053	1.998 ± 0.089	2.065 ± 0.078
Tro/RC	1.725 ± 0.053	1.668 ± 0.058	1.720 ± 0.065
Dio/CSo	69.780 ± 3.870	75.377 ± 7.896	77.787 ± 6.59
Eto/RC	0.964 ± 0.066	0.982 ± 0.043	0.983 ± 0.03
PI_ABS_	3.186 ± 0.598	3.772 ± 0.888	3.237 ± 0.264

## Data Availability

The data presented are available in this manuscript and Appendix A.

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
