# Peer review of "Water and Nutrient Recovery for Cucumber Hydroponic Cultivation in Simultaneous Biological Treatment of Urine and Grey Water"

_plants, 2023, doi:10.3390/plants12061286_

Round 1

Reviewer 1 Report

The article “Water and nutrients recovery for cucumber hydroponic cultivation in simultaneous biological treatment of urine and grey water” reports an interesting investigation aimed at using nitrified urine and grey water as fertilizers for the hydroponic cultivation of cucumber. The work is novel, well-planned, and of sure interest to journal readers, as the general topic has outstanding importance. However, there are some aspects to be fixed before the final publication, mostly related to minor aspects. Thus, I suggest a minor revision which should carefully consider the following points:

1.      Abstract: please define all acronyms at their first appearance in the text (e.g., APX). The same throughout the whole manuscript.

2.      Introduction: recently, the concept of wastewater reuse in agriculture through fertigation practices has raised in popularity worldwide. Comprehensive reviews are provided in 10.3390/su9101734 and 10.1016/j.envpol.2021.118755; they should be considered to briefly depict the general framework of wastewater reuse practices.

3.      Introduction: regarding salinity, the sodium absorption ratio (SAR) parameter should be introduced as well. Also, some quantitative data should be reported in the section (e.g., which are the concentrations of nutrients and toxic elements that are respectively necessary or dangerous to plants?).

4.      Materials and methods, line 545: how many times was nitrified urine diluted? This is not clear.

5.      Material and methods: I would split the section into 2 sub-sections, one related to the cultivation conditions and the other to the characterization methods. Also, I would add a small Table summarizing the significance of each monitored parameter, especially those that are not familiar to all readers (e.g., catalase and ascorbate peroxidase).

6.      Materials and methods: some details about the nitrification process of urine should be mentioned.

7.      Lines 582-584: is there a reference for these empirical equations?

8.      Line 596: I suppose you mean “15000 rpm”. The same in line 599.

9.      Table 1S: the decimal separator should be “.”, not “,”.

10.   Results: the section is well-written, clear, and concise. No significant modifications are required.

11.   Discussion: the section is very long if compared to the length of the other sections. I would recommend cutting it by 20-30%, only focusing on the key points to be delivered to the reader.

12.   Conclusions: some more details about future studies needed on the topic could be added, besides what stated in lines 653-655.

13.   English language, despite being generally good, should be further refined to reach a high-quality standard for publication.

14.   Maybe the supplementary material could be included in the main text, being made only of 2 Tables and 1 Figure.

Reviewer 2 Report

The study results have contributed to scientific debate. Since recycling of urine and grey water for agricultural use can help reduce the amount of chemical fertilizer required, there may be environmental benefits. Using recycled water can also help to conserve water resources, especially in dry areas. The findings of this study could be a significant step toward developing a more sustainable and environmentally friendly water system in modern agriculture.

In this context, I would recommend accepting this article after minor modifications. 

Title

I would suggest modifying the title something like Effects of recycled urine and greywater on cucumber growth….

Abstract

L 19: Rewrite as ” Water and nutrient deficiencies in the soil are becoming a serious threat to crop production.”

L 20: This is confusing sentence “recoveries of water from wastewater” please rewrite it what do you mean by recoveries of water “freshwater” or “processed water”?

L 31: and proline content in a leaves or in the leaves? revise it

Introduction

L 49: content of sodium (Na)? and in L 144 replace sodium with Na.

L 62: replace nitrogen with N and similarly P for phosphorus check throughout the manuscript.

L 69-70: Please mention the name of plants tested already by surfactants.

L 71: Approximately how much concentrations is toxic? And what do you mean by important limitations?

L 73: Please mention some surfactants name.

L 76: Please replace “soilless hydroponic systems” either with “hydroponic systems” or “soilless systems”. Check throughout the manuscript.

L90-91: I would suggest mentioning recent studies on short-rotation crops (https://doi.org/10.1016/j.scitotenv.2021.152209), grass and ornamental flower (https://doi.org/10.1016/j.jenvman.2023.117339) carried out in hydroponics for wastewater remediation. 

L 118-119: remove ”all” and cite references from previous experiment.

Results

The authors have denoted "N" for the number of samples in the tables and figures and captions. I guess it should be n=5?  Please check!

Discussion

Please pay attention to reducing the discussion text in a concise manner.

L311: Please revise this sentence “there has been an increasing amount of discussion regarding the deepening of water deficits”.

L316: By reusing grey water, we can reduce the amount of wastewater and freshwater?

L 319: what do you mean by clean fresh water?

L 329: Delete this sentence “We conducted this study on cucumber plants” as there is no relation to the previous or following sentence.

L 405: Revise this sentence “Under salt stress, a higher accumulation of Na+ in the leaves and roots” as “Salt stress causes an increase in the accumulation of Na+ in the leaves and roots.”

L 406: silenced plants?
